# Class-Weighted Evaluation Metrics for Imbalanced Data Classification

## Abstract

Class distribution skews in imbalanced datasets may lead to models with prediction bias towards majority classes, making fair assessment of classifiers a challenging task. Balanced Accuracy is a popular metric used to evaluate a classifier's prediction performance under such scenarios. However, this metric falls short when classes vary in importance, especially when class importance is skewed differently from class cardinality distributions. In this paper, we propose a simple and general-purpose evaluation framework for imbalanced data classification that is sensitive to arbitrary skews in class cardinalities and importances. Experiments with several state-of-the-art classifiers tested on real-world datasets and benchmarks from two different domains show that our new framework is more effective than Balanced Accuracy – not only in evaluating and ranking model predictions, but also in training the models themselves.

## 1 Introduction

For a broad range of machine learning (ML) tasks, predictive modeling in the presence of *imbalanced datasets* – those with severe distribution skews – has been a long-standing problem (He & Garcia, 2009; Sun et al., 2009; He & Ma, 2013; Branco et al., 2016; Hilario et al., 2018; Johnson & Khoshgoftaar, 2019). Imbalanced training datasets lead to models with prediction bias towards majority classes, which in turn results in misclassification of the underrepresented ones. Yet, those minority classes often are the ones that correspond to the most important events of interest (e.g., errors in system logs (Zhu et al., 2019), infected patients in medical diagnosis (Cohen et al., 2006), fraud in financial transactions (Makki et al., 2019)). While there is often an inverse correlation between the class cardinalities and their importance (i.e., rare classes are more important than others), the core problem here is the mismatch between the way these two distributions are skewed: the $i^{th}$ most common class is not necessarily the $i^{th}$ most important class (see Figure 1a for an illustration). In fact, rarity is one of many potential criteria that can determine the importance of a class, which is usually positively correlated with the costs or risks involved in its misprediction. Ignoring these criteria when dealing with imbalanced data classification may have detrimental consequences.

Consider automatic classification of messages in system event logs as an example (Zhu et al., 2019). An *event log* is a temporal sequence of event messages that have transpired for a given software system (e.g., operating systems, cyber-physical systems) over a certain period of time. Event logs are particularly useful after a system has been deployed. These logs can provide the DevOps teams with information and insights about errors outside of the testing environment, thereby improving their ability to debug and improve the quality of the system. There is typically an inverse correlation between the stability/maturity of a system and the frequency of the errors it produces in its event log. Furthermore, the message types that appear least frequently in an event log are usually the ones with the greatest importance. A concrete example of this was a rare anomaly in Uber's self-driving car that led to the death of a pedestrian, since the system flagged it as a false positive (Efrati, 2018). If this event had not been misclassified and dismissed by the system, the pedestrian death in Arizona may have been avoided.

A plethora of approaches have been proposed for building balanced classifiers (e.g., resampling to balance datasets, imbalanced learning methods, prediction post-processing (Sun et al., 2009; Branco et al., 2016)). A fundamental issue that still remains an open challenge is the lack of a generally-accepted methodology for measuring classification performance. The traditional metrics, which are designed to evaluate average case performance, (e.g., *Accuracy*) are not capable of correctly assess-

(a) Skew in distributions

(b) Example

Figure 1: Skew in distributions of Class Cardinalities or Class Importance, and the potential mismatch between these two distributions render Accuracy and Balanced Accuracy metrics unusable in general multi-class prediction problems.

ing the results in presence of arbitrary skew mismatches between class cardinalities and importances. On the other hand, metrics specifically proposed for imbalanced learning are either domain-specific, do not easily generalize beyond two classes, or can not support varying class importance (e.g., *Balanced Accuracy*) (Japkowicz, 2013).

Let us illustrate the problem with the simple example depicted in Figure 1b. The test dataset consists of 100 data items from 3 classes (A, B, C). The greatest majority of the items belong to class C (70/100), but class B (20/100) has the greatest importance (0.7/1.0). In other words, Cardinality and Importance are both non-uniform and in favor of different classes (i.e., falls in the top-right quadrant of Figure 1a). The confusion matrix on the right-hand side shows the results from a classifier that was run against this test dataset. Unsurprisingly, the classifier performed the best in labeling the majority class C (60/70 correct predictions). When this result is evaluated using the traditional *Accuracy* metric, neither Class Cardinality nor Class Importance is taken into account. If *Balanced Accuracy* is used instead, we observe the degrading impact of the skew in Class Cardinality ($0.38 < 0.65$), but Class Importance is still not accounted for. This example demonstrates the need for a new evaluation approach that is both sensitive to Cardinality and Importance skew, as well as any arbitrary correlations between them. This is especially critical for ensuring a fair assessment of results, when comparing across multiple classifiers or problem instances.

Our goal in this paper is to design an evaluation framework for imbalanced data classification, which can be reliably used to measure, compare, train, and tune classifier performance in a way that is sensitive to non-uniform class importance. We identify two key design principles for such a framework:

- *Simplicity:* It should be *intuitive* and *easy* to use and interpret.
- *Generality:* It should be general-purpose, i.e., (i) *extensible* to an arbitrary number of classes and (ii) *customizable* to any application domain.

To meet the first design goal, we focus on scalar metrics such as *Accuracy* (as opposed to graphical metrics such as ROC curves), as they are simpler, more commonly used, and scale well with increasing numbers of classes and models. To meet the second design goal, we target the more general $n$-ary classification problems (as opposed to binary), as well as providing the capability to flexibly adjust class weights to capture non-uniform importance criteria that may vary across application domains. Note that we primarily focus on *Accuracy* as our base scalar metric in this paper, as it is seen as the de facto metric for classification problems (Sci). However, our framework is general enough to be extended to other scalar metrics, such as *Precision* and *Recall*. Similarly, while we deeply examine two applications (log parsing and sentiment analysis) in this work, our framework in principle is generally applicable to any domain with imbalanced class and importance distributions.

In the rest of this paper, we first provide a brief overview of related work in Section 2. Section 3 presents our new, class-weighted evaluation framework. In Section 4, we show the practical utility of our framework by applying it over three log parsing systems (Drain (He et al., 2017), MoLFI (Messaoudi et al., 2018), Spell (Du & Li, 2016; 2018)) using four real-world benchmarks (Zhu et al., 2019), as well as over a variety of deep learning models developed for sentiment analysis on a customer reviews dataset from Amazon (Ni et al., 2019). Finally, we conclude the paper with a brief discussion of future directions.

## 2 RELATED WORK

**Imbalanced Data Classification.** Imbalanced data is prevalent in almost every domain (Cohen et al., 2006; Batuwita & Palade, 2012; Makki et al., 2019). The growing adoption of ML models in diverse application domains has led to a surge in imbalanced data classification research (He & Garcia, 2009; Sun et al., 2009; He & Ma, 2013; Branco et al., 2016; Hilario et al., 2018; Johnson & Khoshgoftaar, 2019). While the proposed techniques widely vary, they fall under four basic categories: pre-processing training data to establish balance via sampling-based techniques (e.g., (Estabrooks et al., 2004; Blaszczynski & Stefanowski, 2015)), building custom learning techniques for imbalanced training data (e.g., (Joshi et al., 2001; Castro & de Pádua Braga, 2013)), post-processing predictions from an imbalanced model (e.g., (Maloof, 2003)), and their hybrids (e.g., (Estabrooks & Japkowicz, 2001)). In this paper, we do not propose a new imbalanced learning technique, but rather a general-purpose performance evaluation framework that could be used in the training and/or testing of models for any technique. Section 4 demonstrates the practical utility of our framework for a variety of ML models from two different application domains.

**Evaluation Metrics.** Traditional metrics for evaluating prediction performance such as Accuracy, Sensitivity/Specificity (and their combination G-mean), Precision/Recall (and their combination F-Score) were not designed with imbalanced data issues in mind (Japkowicz, 2013). Further, most of these were originally intended for binary classification problems. To extend them to multi-class problems, macro-averaging (i.e., arithmetic mean over individual measurements of each class) is used. Macro-averaging treats each class equally (Branco et al., 2016). Balanced Accuracy is also one such popular averaging-based approach. There are also probabilistic evaluation approaches that extend Balanced Accuracy with Bayesian inference techniques for both binary and multi-class problems (Brodersen et al., 2010; Carrillo et al., 2014). Closer to our work, Cohen et al. (2006) introduced the notion of class weights, yet in the specific context of Sensitivity/Specificity for binary classification in the medical domain. Similarly, Batuwita & Palade (2012) proposed extensions to G-mean for the bio-informatics domain. In addition to these scalar (a.k.a., threshold) metrics, graphical (a.k.a., ranking) evaluation methods such as Receiver Operating Characteristic (ROC) curves or Precision-Recall (PR) curves (and the Area Under the Curve (AUC) for such curves) as well as their extensions to imbalanced data / multi-class problems were also investigated (Weng & Poon, 2008; Japkowicz, 2013). While these methods provide more detailed insights into the operational space of classifiers as a whole, they do not easily scale well with use in problems with a large number of classes (Branco et al., 2016).

## 3 CLASS-WEIGHTED EVALUATION FRAMEWORK

In this section, we present our new evaluation framework designed to measure model accuracy for multi-class classification problems in presence of arbitrary skews among class distributions and/or importances. Our framework builds on and extends commonly used scalar / threshold metrics such as *Accuracy*. These metrics were originally designed for binary classification problems, where there is typically more emphasis on one class (the positive class, e.g., anomalies). To adopt them to multi-class problems where there is no such single-class emphasis, each class' metric can be computed separately and then an overall aggregation (i.e., arithmetic mean) can be performed. For example, *Accuracy* has been extended to *BalancedAccuracy* by following this approach. In our framework, we follow a similar aggregation strategy, however, we do it in a more generalized way that allows custom class weights to capture class importance. Furthermore, these class weights can be based on any importance criteria such as rarity, cost, risk, expected benefits, and possibly a hybrid of multiple such criteria. Therefore, it is critical to provide a flexible formulation that allows users or domain experts to adjust the weights as needed by their problem instance. In what follows, we present our new class-weighted evaluation framework in a top-down fashion. Using the basic notation summarized in Table 1, we first formulate the general framework, and then we describe how this framework can be customized to different importance criteria scenarios by specializing the weights in a principled manner. For ease of exposition, we first focus on *Accuracy* as the underlying performance metric, and then we discuss how our approach can be adopted to other similar metrics.

### 3.1 WEIGHTED BALANCED ACCURACY (WBA)

Suppose we are given a test dataset with $N$ data items in it, each of which belongs to one of $C$ distinct classes. Furthermore, each class $i$ contains $n_i$ of the data items in this dataset. Thus:

Table 1: Notation

| Notation | Description |
|----------|-------------|
| $N$ | total number of data items |
| $C$ | total number of data item classes |
| $M$ | number of importance criteria |
| $n_i$ | true number of data items in class $i$ |
| $p_i$ | correctly predicted number of data items in class $i$ |
| $f_i$ | relative frequency of class $i$ |
| $w_i$ | relative weight of class $i$ |
| $u_i$ | relative user-defined importance of class $i$ |
| $r_i$ | relative rarity of class $i$ |
| $m_{i,j}$ | relative weight of class $i$ for importance criteria $j$ |
| $Accuracy_i$ | Accuracy of class $i$ |

$$N = \sum_{i=1}^{C} n_i \tag{1}$$

The relative frequency of each class $i$ in the whole dataset is:

$$f_i = \frac{n_i}{N} \tag{2}$$

Assume a classifier that makes a prediction about the class label of each data item in the test dataset, and manages to correctly predict $p_i$ out of $n_i$ labels for a given class $i$, where $p_i \leq n_i$. Then, the total number of correct predictions out of all the predictions gives us the overall $Accuracy$ of the classifier as follows:

$$Accuracy = \frac{\sum_{i=1}^{C} p_i}{N} \tag{3}$$

The classifier's $Accuracy_i$ for a given class $i$ (a.k.a., per-class $Recall$ score) can be computed as:

$$Accuracy_i = \frac{p_i}{n_i} \tag{4}$$

$BalancedAccuracy$ is the macro-average of these per-class $Accuracy$ measurements over all classes in the dataset:

$$BalancedAccuracy = \frac{1}{C} \times \sum_{i=1}^{C} Accuracy_i \tag{5}$$

The formulation so far represents the state of the art in how prediction accuracy is evaluated for results of multi-class classifiers in presence of imbalanced datasets (i.e., those where $f_i$ are not even). Note that, for balanced datasets (i.e., $\forall i, n_i = N/C$ and $f_i = 1/C$), $BalancedAccuracy = Accuracy$; whereas for imbalanced datasets, $BalancedAccuracy$ ensures that the prediction accuracy is not inflated due to high-frequency classes' results dominating over the others'. $BalancedAccuracy$ works well as long as each class is of the same importance, since it is the simple arithmetic mean across per-class accuracy measurements of all classes (i.e., each class' accuracy contributes evenly to the overall accuracy). As we discussed with examples in previous sections, in many real-world classification problems, this assumption does not hold. Rather, classifiers must be rewarded higher scores for their prediction performance on more important classes. In order to capture this requirement, we generalize $BalancedAccuracy$ into $WeightedBalancedAccuracy$ by extending it with per-class importance weights $w_i$ as follows:

$$WeightedBalancedAccuracy = \sum_{i=1}^{C} w_i \times Accuracy_i \tag{6}$$

The extension above enables us to capture both skews / imbalances in class cardinalities as well as importances (i.e., the complete design space in Figure 1a). This is a general formulation that can

support any importance criteria for weights as long as $0 \leq w_i \leq 1$ $and$ $\sum_{i=1}^{C} w_i = 1$. In the next section, we present how weights can be customized for different types of importance criteria.

## 3.2 WEIGHT CUSTOMIZATION

In a multi-class problem, not only may the classes carry different importance weights, but also the criteria of importance may vary from one problem or domain to another. In this section, we discuss several types of criteria that we think are commonly seen across many application domains. Note that this is not meant to be an exhaustive list, but it provides examples as well as templates that can be easily tailored to different problems.

**Importance criteria = User-defined.** This is the most general and flexible form of importance criteria. The application designer or domain expert specifies the relative weight of each class based on some application-specific criteria. As an example, the problem might be about classifying images of different types of objects in highway traffic and the user gives higher importance to correct recognition of certain objects of interest (e.g., pedestrians, bikes, animals, etc). We express user-defined relative weight of a class $i$ with $u_i$, which is simply used as $w_i$ in Equation 6.

$$w_i = u_i \tag{7}$$

**Importance criteria = Rarity.** It is often the case that the rarer something is, the more noteworthy or valuable it is. In multi-class problems, this corresponds to the case when importance of a class $i$ is inversely correlated with its relative frequency of occurrence ($f_i$) in the dataset. For example, in system log monitoring, log messages for more rarely occurring errors or exceptions (e.g., denial of service attack) are typically of higher importance. Therefore, a classifier that performs well on detecting such messages must be rewarded accordingly. In our framework, we capture rarity using weights that are based on normalized inverse class frequencies formulated as follows:

$$w_i = r_i = \frac{1}{f_i \times \sum_{j=1}^{C} \frac{1}{f_j}} \tag{8}$$

**Multiple importance criteria.** In some classification problems, importance of a class depends on multiple different criteria (e.g., both rarity and a user-defined criteria). To express class weights in such scenarios, we can leverage techniques from multi-criteria decision making and multi-objective optimization (Triantaphyllou, 2000; Helff et al., 2016). One of the most basic methods is using normalized weighted sums based on composite weights (Helff et al., 2016). Composite weights can be computed either in additive or multiplicative form (Tofallis, 2014). The multiplicative approach tends to promote weight combinations that are uniformly higher across all criteria, and as such is found to be a more preferred approach in application scenarios similar to ours (Helff et al., 2016; Tofallis, 2014). While we present this approach here, in principle, other approaches from multi-criteria decision making theory could also be used together with our framework. Given $M$ different criteria with $m_{i,j}$ denoting the relative weight of class $i$ for criteria $j$, we can compute the composite weight of a class $i$ as follows:

$$w_i = \frac{\prod_{j=1}^{M} m_{i,j}}{\sum_{k=1}^{C} \prod_{j=1}^{M} m_{k,j}} \tag{9}$$

For example, if we had two criteria, rarity $r$ and user-defined $u$ with weights $r_i$ and $u_i$ for each class $i$, respectively, then the composite weight for class $i$ would be $w_i = \frac{r_i \times u_i}{\sum_{j=1}^{C} r_j \times u_j}$.

**Partially-defined importance criteria.** One commonly expected scenario (especially in those classification problems where the number of classes $C$ can be very large) is that not all of the class importance weights might be supplied by the user. For example, in a sentiment analysis use case, the user supplies the weights for all the negative classes, and leaves the others unspecified. Our framework can support such cases by automatically assigning weights to the unspecified classes. The default approach is to distribute the remaining portion of weights evenly across all unspecified classes: (1 - total weights specified) / (number of unspecified classes). If the user prefers an alternative approach (e.g., distribute the remainder based on rarity of the unspecified classes), this can also be easily supported by our framework.

### 3.3 METRIC CUSTOMIZATION

The class-weighted evaluation framework presented above focused on the popular $Accuracy$ metric as the underlying metric of prediction performance. However, our framework follows a general structure based on the idea of *weighted macro-averaging with customizable weights*, which can essentially be used with any performance metric that can be computed on the basis of a class. For example, the macro-averaging approaches that are already being used for $Precision$, $Recall$, and their combination *F-Score* could easily be extended with our customizable weighing approach by replacing $Accuracy$ in our formulas with one of these metrics.

## 4 EXPERIMENTAL ANALYSIS

In this section, we present an experimental analysis of the WBA metric for two application domains. Our primary goal is to demonstrate the value of WBA compared to other standard metrics when evaluating ML models over real-world imbalanced data classification problems. As we will show, often times a traditional metric like $Accuracy$ or $BalancedAccuracy$ will make classifier $A$ seem preferable to classifier $B$, when in reality classifier $B$ is superior. In addition, we also provide a brief analysis of how WBA can positively impact, not only the testing of models, but also their *training*. Further details about this experimental study (including code, data, and examples) can be found in the supplementary material and in Appendix A.

### 4.1 USE CASE 1: LEARNED LOG PARSING

ML-based log parsers are tools that are designed to automatically learn the structure of event logs generated by hardware and software systems to properly categorize them into event classes (e.g., different error types). In our first study, we used WBA to evaluate 3 state-of-the-art log systems: Drain, Spell, and MoLFI (Du & Li, 2016; He et al., 2017; Messaoudi et al., 2018). We start by providing an abbreviated description of our experimental setup.

**Log Parsing Systems.** *Drain* is a rule-based, online log parsing system that encodes the parsing rules in a parse tree with fixed depth (He et al., 2017). It performs a pre-processing step for each new log message using regular expressions created by experts with domain knowledge. *Spell*, like Drain, is also rule-based; it principally uses the *longest common subsequence* (LCS) to find new classes of log messages (Du & Li, 2016). It parses messages in an online fashion by creating objects for each message type containing information about LCS. Finally, *MoLFI* casts the log parsing as a multi-objective optimization problem and provides a solution based on genetic programming (Messaoudi et al., 2018).

**Datasets.** We test each aforementioned log message classification system with four real-world datasets taken from a public benchmark (Zhu et al., 2019). Each dataset has 2000 log instances randomly sampled from a larger dataset. The *macOS* dataset contains raw log data generated by the macOS operating system (341 log classes, 237 *infrequent classes* (i.e., those that have fewer occurrences in the dataset than the average number of messages per class), and an average class frequency of 5). The *BlueGene/L (BGL)* dataset is a collection of logs from the BlueGene/L supercomputer system (120 log classes, 101 infrequent classes, and an average class frequency of 16). The *Android* dataset consists of logs from the Android mobile operating system (Zhu et al., 2019) (166 log classes, 127 infrequent classes, and an average class frequency of 16). Finally, the *HDFS* dataset consists of log data collected from the Hadoop Distributed File System (14 log classes, 8 infrequent classes, and an average class frequency of 142). Overall, the first three datasets are highly skewed in class frequencies, whereas the HDFS dataset is relatively much less skewed (see Appendix A).

**Results.** For Drain, Spell, and MoLFI, traditional metrics of $Precision$, $Recall$, $F1\text{-}Score$, and $Accuracy$ (named *Parsing Accuracy* in the original papers) were used for training and testing classification performance. None of these metrics are class-sensitive, while in log parsing, messages have in fact varying importance across the classes. The importance criteria is rarity: the more rare an error message is, the more important it is to correctly classify this message. To capture this, we configure the WBA to $WBA_{rarity}$, which automatically assigns weights to WBA based on the dataset classes' *inverse frequencies*, as described in Section 3.2. Then we evaluate the test results from the 3 parsers over 4 datasets using $WBA_{rarity}$ and compare against traditional metrics in two categories: class-insensitive and class-sensitive, as shown in Figure 2.

$WBA_{rarity}$ *vs. Class-insensitive Metrics:* The class-insensitive metrics (specifically, F1-Score and

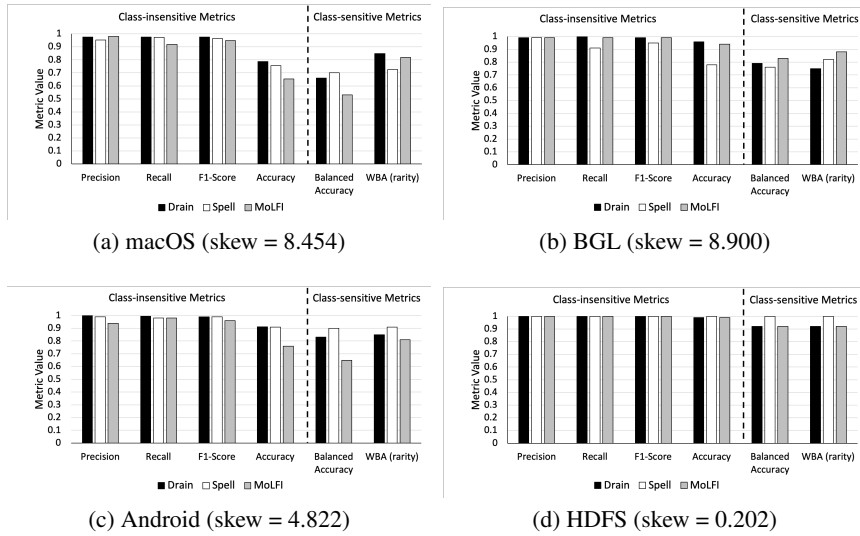

Figure 2: WBA vs. class-insensitive & class-sensitive metrics for log parsing: F1-Score & Accuracy agree in all. BA & WBA agree in (c) & (d) only. WBA disagrees with class-insensitive in all.

Accuracy) agree on how to rank the classification performance of the 3 parsers across all the datasets (for macOS and Android, Drain > Spell > MoLFI; for BGL, Drain > MoLFI > Spell; for HDFS, all perform similarly). Since $WBA_{rarity}$ is sensitive to classes' data distribution and importance skews, it makes a completely different judgement. Furthermore, it ranks the techniques differently for each dataset (Drain > MoLFI > Spell in macOS; Drain > Spell > MoLFI for BGL; Spell > Drain > MoLFI for Android; and for HDFS, Spell > Drain and MoLFI). This result validates that $WBA_{rarity}$ provides a more sensitive tool for assessing classification performance.

$WBA_{rarity}$ vs. Balanced Accuracy (BA): As discussed earlier, BA is class-sensitive, but only to distribution imbalance. We can observe the difference between BA and WBA in Figure 2. In macOS and BGL, where the skew is the highest and rarity is more pronounced, the two metrics completely disagree in how they rank the parsers. In contrast, for Android and HDFS, where the skew is lower, there is an overall agreement, although the separation in metric values slightly differ. Of particular importance is the difference seen in Figure 2a. We observe that the best performing model is Spell when scored by BA, and Drain when scored by $WBA_{rarity}$. The reason for this difference is due to Spell's and Drain's differences in their ability to correctly classify infrequent classes, i.e., those that represent failures and errors that require the most immediate response.

### 4.2 USE CASE 2: SENTIMENT ANALYSIS

In social media and other user-facing domains like e-commerce sites, it is often useful to understand the view or feelings ("sentiments") associated with users' behavior or preferences. In the second part of our experimental study, we apply WBA in the context of such a sentiment analysis use case, which involves analyzing text-based product reviews from Amazon's e-commerce websites.

**Dataset.** The dataset consists of customer's reviews and ratings, which we got from Consumer Reviews of Amazon products (Ni et al., 2019). The task is to classify the reviews into 5 classes (with 1 being the lowest and 5 being the highest rating a product can get in a review), where ratings constitute the ground truth class labels. There is high class imbalance in this dataset (skew=2.140). As shown in the Frequency column of Table 2, Class 5 with the highest customer rating clearly dominates compared to the other classes. It is known that the distribution of customer review ratings is typically imbalanced and generally follow a J-shaped distribution (Mudambi & Schuff, 2010; Pavlou & Dimoka, 2006).

**Sentiment Analysis Models.** We compare 4 types of recurrent neural networks (RNN), all consisting of an embedding layer with pre-trained word embeddings from (Pennington et al., 2014) followed by a recurrent layer from PyTorch (Subramanian, 2018): RNN, LSTM, GRU, BiLSTM. The hidden state output from the last time step of these are passed to a fully-connected layer with

Table 2: Amazon per-class breakdown: Frequencies are highly skewed (skew=2.140); $Accuracy_i$ in each model when both trained+tested with user-defined weights $w_i$ (same weights as in Figure 3a).

| Class | Frequency ($f_i$) | Weights ($w_i$) | LSTM | RNN | GRU | BiLSTM |
|---|---|---|---|---|---|---|
| 1 | 0.092 | 0.7 | 0.19 | 0.04 | 0.16 | 0.17 |
| 2 | 0.052 | 0 | 0 | 0 | 0 | 0 |
| 3 | 0.075 | 0 | 0 | 0 | 0 | 0 |
| 4 | 0.142 | 0 | 0 | 0 | 0 | 0 |
| 5 | 0.639 | 0.3 | 0.81 | 0.96 | 0.84 | 0.83 |

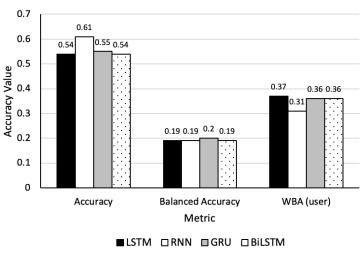

(a) WBA vs. Other Metrics (Train+Test)

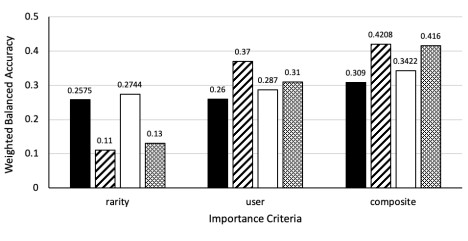

(b) WBA: Test vs. Train+Test

Figure 3: Amazon results

input of 256 neurons and output from 5 neurons.

**Results.** For this use case, we first worked with a user-defined importance criteria borrowed from published studies suggesting that extreme review ratings (classes 1 and 5) carry more importance (Mudambi & Schuff, 2010; Pavlou & Dimoka, 2006). Thus, we set the weights as shown in Table 2 (shown as WBA(user) or user in Figure 3).

*WBA vs. Other Accuracy Metrics:* First, we compare WBA(user) with Accuracy and BalancedAccuracy (BA) when used as a metric for both training and testing of the 4 DNN models (Figure 3a). We make a few observations: (i) The class-insensitive Accuracy showcases the imbalance problem in classification, as it favors the RNN model which is heavily biased by the majority class (see $Accuracy_i$ for RNN in Table 2 where class 5 scores 0.96). (ii) The frequency-sensitive BA metric finds all models perform similarly. WBA(user), in contrast, identifies LSTM as the best model. Indeed, Table 2 confirms that LSTM performs relatively the best in predicting the most important class, class 1 (0.19 accuracy). Overall, we find that WBA is capable of capturing importance skews, even when the frequency skew can be high and biased towards less important classes.

*Impact of WBA in Model Training:* Next we explore the use of WBA not only in model evaluation, but also in training. We focus on two models (LSTM and RNN), and apply WBA only during testing vs. to both training (by modifying loss functions of DNNs to capture class importance weights) and testing. Intuitively, if a model is trained being aware of the importance weights, then it should also perform well when tested against the same criteria. To test this hypothesis, we repeated the experiment for 3 alternative importance criteria: (i) rarity ($w_1 = 0.209$, $w_2 = 0.368$, $w_3 = 0.255$, $w_4 = 0.136$, $w_5 = 0.030$), (ii) user-defined (i.e., with weights in Table 2), and (iii) composite of the two ($w_1 = 0.62$, $w_2 = w_3 = w_4 = 0$, $w_5 = 0.38$). In Figure 3b, we observe: (i) Except for rarity, WBA for both LSTM and RNN improves when integrated into model training. This verifies our intuition, and shows that WBA is a useful metric not only for evaluation, but also for training. (ii) When we zoom into rarity, we see that although class 2 is the most important, per-class accuracy for class 5 is much higher for both LSTM and RNN in the *Test-only* case, because both models are still trained heavily biased towards the majority class (5). (iii) Though rarity by itself is not useful in training, when combined with user importance, it visibly improves the WBA scores. This shows that our multi-criteria composition approach is capable of combining importance criteria as intended.

## 5 CONCLUSION

In this paper, we presented a simple yet general-purpose class-sensitive evaluation framework for imbalanced data classification. Our framework is designed to improve the grading of multi-class classifiers in domains where class importance is not evenly distributed. We provided a modular and extensible formulation that can be easily customized to different importance criteria and metrics. Experiments with two real-world use cases show the value of a metric based on our framework, Weighted Balanced Accuracy (WBA), over existing metrics – in not only evaluating the classifiers' test results more sensitively to importance criteria, but also training them so.

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

# A APPENDIX

In this appendix, we provide details for the experimental study, including data and code. For further information, please see the supplementary material.

## A.1 DETAILS FOR LOG PARSING EXPERIMENTS

For the three log parsing techniques used in Section 4.1 (Drain, Spell, and MoLFI), we used the implementations provided by the LogPAI team:

`https://github.com/logpai/logparser/`

The four datasets used in these experiments (macOS, BGL, Android, and HDFS) came from the benchmarking data also provided by LogPAI:

`https://github.com/logpai/loghub/`

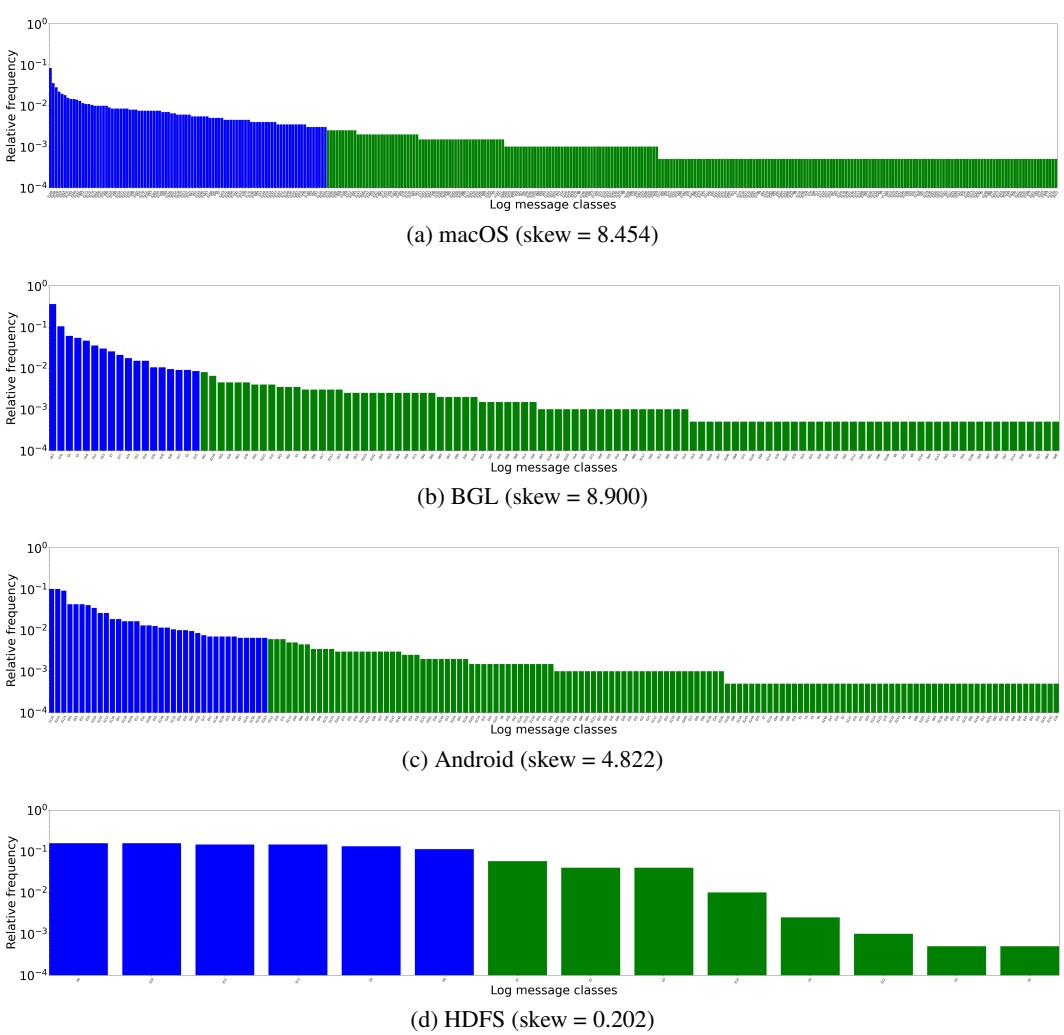

(a) macOS (skew = 8.454)

(b) BGL (skew = 8.900)

(c) Android (skew = 4.822)

(d) HDFS (skew = 0.202)

Figure 4: Histograms showing the relative frequencies of log parsing classes for the four experimental datasets: All graphs have their y-axes in log scale; green bars show the *infrequent classes*.

In Figure 4, we show the histograms for the four log datasets together with their skew values. As defined in the Microsoft Excel Documentation, "*Skewness characterizes the degree of asymmetry of a distribution around its mean. Positive skewness indicates a distribution with an asymmetric tail extending toward more positive values, while negative skewness indicates a distribution with an*

*asymmetric tail extending toward more negative values.*" [1]. In our context, skew provides a good indication for the degree of imbalance in class cardinality distributions – the larger the skew, the larger the degree of class imbalance.

We also provide data files with class labels (true + predicted) and weights (based on rarity as importance criteria) used in generating the experimental data plotted in Figure 2 as part of our WBA-Evaluator tool implementation included in the supplementary material (can be found under the `WBA-Evaluator/examples/LogParsing/` directory).

## A.2 DETAILS FOR SENTIMENT ANALYSIS EXPERIMENTS

For the sentiment analysis experiments of Section 4.2, we used a sample from the Amazon Customer Reviews dataset provided at:

`https://nijianmo.github.io/amazon/index.html`

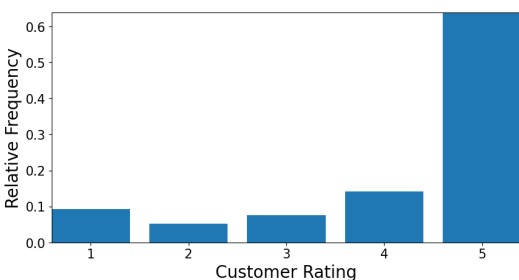

Figure 5: Histogram showing the relative frequencies of the five customer rating classes for the Amazon dataset (skew = 2.140).

In Figure 5, we show the histogram for the Amazon dataset. As described in Section 4.2, we implemented 4 RNN-based classifiers to experiment with this dataset. The code for these classifiers can be found in the supplementary material (under the `AmazonReviewsClassifier/src/` directory) along with a copy of the data (under the `AmazonReviewsClassifier/dataset/` directory).

We also provide the data files with class labels (true + predicted) and weights (user) used in generating the experimental data for LSTM results plotted in Figure 3 and Table 2 as an example. These can be found in our WBA-Evaluator tool implementation included in the supplementary material under the `WBA-Evaluator/examples/Amazon/` directory.

## A.3 THE WBA-EVALUATOR TOOL

In addition to details on our experimental study as described above, we also provide a copy of the WBA-Evaluator tool that implements our customizable, class-weighted evaluation framework described in Section 3. WBA-Evaluator is written in Python and can be found in the supplementary material along with a README that describes how it can be used. In a nutshell, WBA-Evaluator takes as input three files (true class labels, predicted class labels, class weights) and a number of configuration parameters in the form of commandline arguments, and then it generates accuracy scores (BA or WBA) as specified by these arguments. The WBA-Evaluator implementation comes with two subdirectories: `src/` contains the Python source code; `example/` contains all the input files (labels and weights) and scripts in the `scripts/` subfolder to run these. Please see the `README` file for more details. Using this tool, results reported in the paper can be reproduced.

---

[1] `https://support.microsoft.com/en-us/office/skew-function-bdf49d86-b1ef-4804-a046-28eaea69c9fa`

