# OpenReview forum: "Class-Weighted Evaluation Metrics for Imbalanced Data Classification"
_ICLR.cc/2021/Conference — Reject_

### Official Review · AnonReviewer2 · 2020-10-25
**An interesting little idea without full evaluation**

**Rating:** 6
**Confidence:** 4

**Review:**

The paper presents a simple addition to the Balanced Accuracy approach - which the authors refer to as ‘importance’. However, there is nothing in the formulation of this concept which requires that this is an importance and could in fact be any form of weighting. The paper evaluates the new metric - but only agains the Balanced Accuracy metric (which seems quite restrictive).

## Some major comments on the paper:

1) The proposed evaluation metric appears to be to show whether machine learning approach A is actually better than machine learning approach B. As such one can use the metric to give a value which can be used to compare different approaches. However, in order to judge if a particular approach is better or worse than another you need some way of showing that your metric is correct. The paper lacks a demonstration that your approach actually does give a more appropriate ordering of the machine learning approaches.

2) In the abstract you talk about ‘importance’, however, this concept is not clear at this point. Later in the introduction you explain what importance refers to. Some of this in the abstract would help. The abstract also seems to terse making it difficult to follow.

3) The second paragraph in the introduction seems odd. You are claiming in the previous paragraph that importance need not be the inverse of rarity, however, the example in the second paragraph seems to re-enforce the idea of importance being the inverse of rarity. The text on Uber also doesn’t seem to fit the paper at all and could easily be dropped.

4) Figure 1a needs more discussion in the text.

5) Paragraph starting ‘Let us illustrate the problem’ - this talks about a dataset in the most abstract sense. It would seem this is potentially a concocted example without a real dataset behind it. It would be much better to indicate what dataset this is based on - perhaps detailing the scenario in the appendix.

6) The related work on evaluation metrics seems a little short. How do the other proposed approaches compare to your work?

7) “As we discussed with examples in previous sections, in many real-world classification problems,” - you only give one example and the ‘real-world’ case that this refers to is not provided. Stronger evidence is needed to support this statement.

8) Most sentiment analysis approaches are not based on RNNs - can you justify why you used this approach?

9) The conclusions are rather short and say little. You also claim in the introduction that you will discuss future directions but don't.

## Some more minor comments:

- If a majority is the larges group, what is the ‘greatest majority’? Surely it’s just the majority?

- ‘adopt’ -> ‘adapt’

- There is no punctuation around the equations - For example there should be a full-stop after equations 1 and 2.

- As only equation 6 is referred to in the text why are the others numbered?

- Something going odd in the quotes in “ not inflated due to high-frequency classes’ results dominating over
the others’. “

- “must be rewarded higher scores” -> “must be rewarded with higher scores”

- Is equation 7 really needed?

---

### Official Review · AnonReviewer5 · 2020-11-04
**Not enough novelty and originality**

**Rating:** 3
**Confidence:** 5

**Review:**

This paper proposes a simple and general-purpose evaluation framework for imbalanced data classification that is sensitive to arbitrary skews in class cardinalities and importances.

I think the problem this paper deals with is very important and of great interest to the wide range of readers.
In addition, the paper is generally clearly written and easy to follow.

However, I found the novelty and the originality of this paper is not enough for the ICLR standard.
Although I am not aware of the paper that presents exactly the same concept as this paper, I feel the generalization this work presents is too straightforward that new insights, findings, and benefits brought by this paper to the community are very limited.
For example, the micro average (or simply called "Accuracy" in this paper) can be regarded as a special form of equation (6), where weights or importance is given by the relative frequency of each class.
This importance criteria can be regarded as opposite of one of the presented criteria, "Rarity".
There may be some cases where the rarer a class is, the more important the class is as explained in the paper, but there may be other cases where more frequent classes are more important. As such, the concept of including importance of each class into an evaluation metric has been implicitly considered. It is true that the paper provides general form that includes the aforementioned case, but I'm afraid the formulation is not novel enough to bring a new value to the community as I mentioned above.

Another weakness of the paper lies in the experimental analysis.
Overall, the analysis is not convincing enough to verify the benefit of the proposed metric.
For example, at the end of "WBA_rarity vs. Class-insensitive Metrics" in page 7, the authors states "This result validates that WBA_rarity provides a more sensitive tool for assessing classification performance". I do not agree with this statement because it is no surprise that different evaluation metrics give different evaluation results, and this alone is not the ground for the validity of the proposed metric. The same argument can be applied to the "Impact of WBA in Model Training" in page 8. It is natural that a model trained with a specific criteria results in performing well on the criteria. This is again not the ground for the claim "our new framework is more effective than Balanced Accuracy – ... also in training the models themselves."

The pros and cons of this paper can be summarized as follows.


#### Pros
1. The paper deals with practically important topic, and presents a simple, easy and intuitive solution.
1. The paper is clearly written and easy to understand.

#### Cons
1. The novelty and the originality of this paper is not enough that there is only limited benefits to the community, which does not satisfy the ICLR standard.
1. The experimental analysis is not convincing enough to support the usefullness of the proposed metric.

---

### Official Review · AnonReviewer6 · 2020-11-04
**lacks contribution**

**Rating:** 3
**Confidence:** 4

**Review:**

This paper presents a weighted balanced accuracy to evaulate the performance of multi-class classification. Basically, the performance for a multi-class problem can be evaluated by decomposing the original multi-class problem into a number of binary ones based on one-against-rest manner, and then evaulating the performance scores for each of the binary ones using any well-known metric for binary classification, and then, aggregating the performance scores. The main aim of this paper is to present a weighting scheme when aggregating the scores.

I'm inclined to rejection of this paper. Frankly speaking, I don't think this paper has a significant contribution. the weighting schemes to combine binary metric scores to evaulate the performance of multi-class classification have been well-studied, such as macro-averaging, micro-averaging, as well as importance weighting (manual or data-driven e.g. frequency). The proposed idea is just a simple natural extension of balanced accuracy with a weighting scheme. It is nothing new.

---

### Official Review · AnonReviewer4 · 2020-11-09
**Novel insights but more works need to be done**

**Rating:** 4
**Confidence:** 4

**Review:**

Summary:

This paper proposes a new evaluation framework for imbalanced data. Specifically, they introduce an additional weighted term in the formulation of balanced accuracy. By varying the weights, their framework can be adapted to many application domains. Finally, they use two case studies to illustrate the effectiveness of the proposed measure.

Pros:

- The proposed framework is simple and effective.
- The proposed framework can be used in many application domains.


Cons:

I have several comments regarding the experimental results:

- How do the authors compute the “skew” score for each dataset? This term is not clearly explained in the paper.

- Since the weighted terms e.g., $w_i$ play an important role in the proposed framework, more examples should be given to explain how to choose these parameters in different application domains.

- In section 4.2, the authors state “ … by modifying loss functions of DNNs to capture class importance weights…”. It is not very clear to see how to implement this.

- When using WBA in model training, what are the performances for the baseline measures, e.g., precision, recall? Also, why choose these values for $w_i$, e.g., w1=0.209, w2=0.368, and w3=0.255?


Overall, I think the novelty of this work is limited and the experimental results are not very convincing.

---

### Decision · Program_Chairs · 2021-01-07
**Final Decision**

**Decision:**

Reject

**Comment:**

This paper proposes a weighted balanced accuracy metric to evaluate the performance of imbalanced multiclass classification. The metric is based on a one-versus-all decomposition from multi-class to binary, and then aggregating the metrics on the binary classification sub-problems in a weighted manner. The authors hope to argue that the new metric is more flexible for evaluating classifiers in the imbalanced and importance-varying setting.

The reviewers agree that the proposed framework is simple and applicable to an important problem. Somehow the novelty and significance of the work is pretty limited, as many related metrics (e.g. micro/macro-averaged metrics) exist in the literature. The authors are encouraged to think about stronger reasoning on how useful the "new" metric is. The experiments are also not convincing nor complete enough to verify the benefits of the proposed metric.